# Phosphorus-Driven Stem-Biased Allocation: NPK Synergy Optimizes Growth and Physiology in *Dalbergia odorifera* T. C. Chen Seedlings

**DOI:** 10.3390/plants14162545

**Published:** 2025-08-15

**Authors:** Mengwen Zhang, Chuanteng Huang, Ling Lin, Lin Chen, Xiaoli Yang, Xiaona Dong, Jiaming Song, Feifei Chen

**Affiliations:** 1Hainan Academy of Forestry (Hainan Academy of Mangrove), Haikou 571100, China; zaizai_qq@163.com (M.Z.); huangchuanteng@126.com (C.H.); lling008@163.com (L.L.); 13707569339@163.com (L.C.); 18889827255@163.com (X.Y.); dongxiaona@163.com (X.D.); 15035584294@163.com (J.S.); 2The Innovation Platform for Academicians of Hainan Province, Haikou 571100, China; 3Key Laboratory of Tropical Forestry Resources Monitoring and Application of Hainan Province, Haikou 571100, China

**Keywords:** *Dalbergia odorifera*, “3414”fertilization, growth, physiology, cultivation

## Abstract

Valued for furniture, crafts, and medicine, *Dalbergia odorifera* T. C. Chen confronts critically depleted wild populations and slow cultivation growth, necessitating precision nutrient formulation to overcome physiological constraints. Using a ‘3414’ regression design with four levels of N, P, and K, this study identified phosphorus (P) as the most influential nutrient in regulating growth (P > N > K). Maximal growth enhancement occurred under T7 (N2P3K2), with height and basal diameter increments increasing by 239% and 128% versus controls (*p* < 0.05). Both traits exhibited progressive gains with rising P but unimodal responses to N and K, initially increasing then declining. T7 boosted total biomass by 50% (*p* < 0.05) with stem-biased partitioning (stem > root > leaf; 52%, 26%, 22%). Photosynthetic capacity increased significantly under T7 (*p* < 0.05), driven by P-mediated chlorophyll gains (Chla + 70%; Chlb + 75%) and an 82% higher net photosynthetic rate. Metabolic shifts revealed peak soluble sugar in T7 (+139%) and soluble protein in T9 (+226%) (*p* < 0.05), associated primarily with P and K availability, respectively. Correlation networks revealed significant associations among structural growth, photosynthesis, and metabolism. Principal component analysis established T7 as optimal, defining a “medium-N, high-P medium-K” precision fertilization protocol. These findings elucidate a phosphorus-centered regulatory mechanism governing growth in *D. odorifera*, providing a scientific foundation for efficient cultivation.

## 1. Introduction

Tropical precious timber species hold strategic importance for global forest conservation and sustainable resource utilization, with their combined economic and ecological value driving advancements in artificial cultivation research [1]. *Dalbergia odorifera* T. C. Chen (Hainan Huanghuali), a Leguminosae species endemic to Hainan, China, possesses significant economic value due to its durable corrosion-resistant heartwood exhibiting fine figuring and reddish-brown to dark red coloration with distinctive aromatic properties [2,3]. Furthermore, its heartwood serves as a traditional Chinese herbal medicine, valued for properties such as promoting blood circulation, alleviating pain, stopping bleeding, and reducing swelling [4]. However, wild populations have been critically depleted through prolonged overexploitation and habitat degradation, resulting in dual conservation designations: as a National Second-Class Protected Plant in China and as Endangered (EN) on the IUCN Red List, with extant wild specimens below 100 mature individuals [5]. Artificial afforestation has, thus, become the primary approach to addressing the resource supply–demand imbalance. Nevertheless, the species faces significant challenges: a long growth cycle (requiring decades to mature), slow seedling growth, and high sensitivity to environmental stress [6,7]. These factors contribute to a shortage of high-quality seedlings, severely constraining the industrialization of artificial forests [8,9]. Therefore, developing key technologies to regulate *D. odorifera* seedling growth is pivotal for enhancing seedling quality and accelerating resource recovery.

Plant growth and development are regulated by the interplay of genetic factors and environmental conditions, with nutrient supply constituting a central environmental determinant of seedling morphogenesis and physiological metabolism [10,11]. Nitrogen (N), phosphorus (P), and potassium (K), as essential macronutrients, participate in fundamental physiological processes including photosynthesis, energy metabolism, and biosynthesis. Their availability directly governs seedling growth rates and stress tolerance [12,13]. Specifically, nitrogen―a structural component of chlorophyll, proteins, and nucleic acids―is vital for plant function; its deficiency induces chlorosis and growth retardation [14]. Phosphorus, a core constituent of ATP and phospholipids, facilitates energy transfer and signal transduction; P deficiency impairs root development and reproductive processes. Potassium primarily functions as an enzyme cofactor, regulating stomatal aperture and osmotic homeostasis; K deficiency reduces photosynthetic efficiency and stress resilience [15]. Synergistic interactions among these elements are critical for maintaining growth homeostasis. Imbalances in individual element supply or suboptimal ratios may trigger nutrient antagonism, exemplified by elevated nitrogen suppressing phosphorus/potassium uptake, or excess phosphorus immobilizing micronutrients such as iron (Fe) and zinc (Zn) [16,17,18]. Consequently, defining the optimal N:P:K ratio is fundamental to achieving the precision cultivation of high-quality nursery stock.

Nutrient ratio fertilization has been extensively studied in valuable timber species including *Carya illinoinensis* (Wangenh.) K. Koch [19], *Moringa oleifera* Lam. [20], *Catalpa bungei* C. A. Mey [21], and *Pinus massoniana* Lamb. [22]. Evidence suggests that tailoring fertilizer ratios to seedling nutrient demands is crucial for optimizing fertilizer efficiency, as both excessive and insufficient application can lead to resource wastage or nutrient deficiencies [23,24]. Research on species such as *Phoebe bournei* (Hemsl.) Yen C. Yang and *Manglietia glauca* Blume demonstrates that balanced N, P, and K application enhances photosynthetic pigment content, improves leaf light-harvesting capacity, accelerates photosynthetic product synthesis, and promotes seedling height, stem diameter growth, and biomass accumulation [25,26]. Owing to interspecific genetic variation, fertilizer efficacy differs markedly: P predominantly regulates photosynthesis in *M. glauca*, whereas N is the primary regulator in *C. illinoinensis* [25,26,27]. Optimal fertilization rates also vary significantly among species [28,29]. While appropriate nutrient supplementation promotes seedling growth, excessive application may cause phytotoxicity [30]. Despite these advances, research on *D. odorifera* reveals critical knowledge gaps: Existing studies predominantly isolate single-nutrient effects (e.g., nitrogen-driven biomass thresholds [31]), lacking mechanistic insight into NPK interactions. Although preliminary NPK combinations (e.g., N2P3K1 and N3P3K1) suggest potential for optimizing growth or photosynthesis, limited treatment designs fail to quantify multi-factor synergies [32,33]. Most consequentially, the coupled responses of growth (height, diameter), photosynthetic function (net rate, fluorescence), and carbon metabolism (e.g., soluble sugars) remain unresolved, impeding predictive model development.

Given the marked interspecific variation in nutrient responses and the limitations of current *D. odorifera* research, standardized experimental designs that integrate multi-factor interactions are urgently needed. The “3414” design—an incomplete orthogonal regression design with three factors (N, P, K) and four levels—provides unique advantages here, including comprehensive factor inclusion, graded level settings, operational simplicity, and robust analytical methods, which align with the requirements for robust fertilizer decision making [34]. While this design has been successfully applied in species like *P. bournei* and *Anisodus tanguticus* (Maxim.) Pascher to quantify nutrient synergies [26,35], it remains unexplored for *D. odorifera.* Notably, current research on *D. odorifera* disproportionately compartmentalizes key metrics: morphological indicators are often decoupled from photosynthetic efficiency, and carbon allocation trade-offs between source (e.g., fluorescence-derived photochemical yield) and sink (e.g., stem-soluble sugars/proteins) remain unquantified under NPK gradients, gaps that the “3414” design, with its capacity to integrate multi-factor responses, is well suited to address.

To address these gaps, this study utilizes three-year-old *D. odorifera* seedlings in an NPK fertilization experiment employing the “3414” design to systematically (1) quantify individual and interactive effects of N, P, and K supply on seedling morphological development (height, basal diameter) and biomass partitioning; (2) decipher physiological response mechanisms governing photosynthetic pigment dynamics, gas exchange, and carbon–nitrogen metabolic flux across NPK gradients; and (3) establish synergistic linkages among growth–physiology traits and identify the optimal NPK ratio through multi-trait dimensionality reduction analysis. The empirical evidence will advance our mechanistic understanding of nutrient regulation in *D. odorifera*, providing evidence-based fertilization protocols to support sustainable cultivation of this endangered species.

## 2. Results

### 2.1. Regulatory Effects of NPK Fertilization on Seedling Morphogenesis

NPK fertilization significantly influenced morphological development in *Dalbergia odorifera* T. C. Chen seedlings (height and basal diameter increments, *p* < 0.05). All treatments yielded greater height increments than the control (T1), peaking in T7 (+239%) and followed by T6 (+191%); all treatments except T2, T5, T8, and T14 showed significant increases (Figure 1A). Basal diameter increment peaked in T7 (+128%), with T6 (+104%) ranking second; only T6, T7, T9, and T11 differed significantly from T1 (Figure 1B). Univariate analysis revealed single-peak responses to nitrogen (N) and potassium (K) at medium levels, whereas both parameters increased continuously with phosphorus (P) supply, achieving highest values at high P levels. Range analysis established the fertilizer effect ordination as P > N > K (Table 1).

### 2.2. Nutrient-Driven Patterns of Biomass Accumulation and Allocation

Fertilization significantly regulated dry matter accumulation (*p* < 0.05). The highest leaf dry weight (LDW) occurred in T6 (+80% vs. control), while stem (SDW), root (RDW), and total dry weight (TDW) reached maximal accumulation in T7 (+46%, +43%, and +50% vs T1, respectively; Figure 2). Biomass partitioning exhibited a stem-biased pattern (stem: 52% > root: 26% > leaf: 22%). Univariate analysis revealed optimal LDW under medium N, P, and K levels. SDW, RDW, and TDW increased monotonically with P supply, reaching maxima at high P levels; SDW and TDW responded optimally to medium N and low K levels, whereas RDW was highest at medium N and K levels. The fertilizer effects were ranked as follows: LDW—N > P > K; SDW—P > K > N; RDW/TDW—P > N > K (Table 2).

### 2.3. Response of Photosynthetic Pigment Synthesis to NPK Fertilization

Fertilization significantly affected photosynthetic pigment content (*p* < 0.05). Chlorophyll a (Chla) attained its highest level in T7 (+68.02% vs. T1), followed by T6 (+54%) and T9 (+51%) (Figure 3A). Chlorophyll b (Chlb) was maximal in T7 (+75%), with T6 (+68%) and T11 (+66%) showing the next greatest increases (Figure 3B). Total chlorophyll (TChl) attained its maximum in T7 (+70%), and then T6 (+58%) and T11 (+52%) (Figure 3C). Pigments showed optimal responses to medium N and K levels but increased monotonically with P supply, achieving maxima at high P levels. Fertilizer efficacy was ranked as follows: Chla—P > N > K; Chlb—N > P > K; TChl—P > N > K (Table 3).

### 2.4. Synergistic Response of Gas Exchange Parameters to Nutrient Levels

Fertilization significantly altered gas exchange parameters (*p* < 0.05). The net photosynthetic rate (P_n_) achieved its maximum in T7 (+82% vs. T1; Figure 4A). Stomatal conductance (g_s_) increased significantly across all treatments except T4. T9 exhibited the highest g_s_ value, though it was not statistically different from T6, T7, and T10 (Figure 4B). Transpiration rate (T_r_) increased significantly in all treatments except T2, T4, and T8, with the highest value in T7 (Figure 4C). Intercellular CO_2_ concentration (C_i_) was significantly lower than T1 in all treatments except T2, T4, and T8; T3 recorded the minimum value (Figure 4D). Univariate analysis revealed optimal responses of P_n_, g_s_, and T_r_ to medium N and low K levels, while all three increased monotonically with P supply. C_i_ decreased progressively with N, P, and K application. The fertilizer effects were ranked as follows: P_n_—P > N > K; g_s_—P > K > N; T_r_—P > K > N; C_i_—P > N > K (Table 4).

### 2.5. Fertilization Regulation of Carbon and Nitrogen Metabolism Products

Fertilization significantly influenced soluble sugar content (SSC) and soluble protein content (SPC) in *D. odorifera* seedlings (*p* < 0.05). All treatments exceeded the control (T1) in SSC, with T7 and T9 showing highest increases of 139% and 134%, respectively (Figure 5A). Similarly, SPC was elevated in all treatments except T2, T4, and T8; T9 and T7 exhibited maximal SPC levels, increasing by 226% and 192% versus T1 (Figure 5B). Univariate analysis revealed monotonic increases in SSC with N and P supply (reaching maxima at high levels), but an optimal response to K at low levels. SPC increased steadily with P (peaking at high P), while showing optimal responses to N and K (at medium N and low K levels, respectively). The fertilizer effects were ranked as follows: SSC—P > K > N; SPC—K > N > P (Table 5).

### 2.6. Quantitative Screening of Optimal Fertilizer Ratios

#### 2.6.1. Interaction Network of Growth–Physiological Indicators

A correlation network analysis was performed using Pearson’s method adjusted with Bonferroni correction (*α* = 0.01), revealing tight functional coupling among 15 growth–physiological traits in *D. odorifera* seedlings (Figure 6). Key relationships included:(1)Biomass coordination: BDI exhibited strong positive correlations (*r* = 0.92–0.96) with all biomass components (LDW, SDW, RDW, TDW). TDW functioned as a central hub, showing robust correlations (*r* = 0.90–0.98) with each organ-specific biomass measure (all correlations *p* < 0.01).(2)Photosynthetic and metabolic integration: Photosynthetic pigments (Chla, Chlb, TChl) demonstrated high intercorrelation (*r* = 0.92–0.99). Chla and TChl correlated positively with T_r_ (Chla-Tr: *r* = 0.91; TChl-T_r_: *r* = 0.90) and SPC (Chla-SPC: *r* = 0.93; TChl-SPC: *r* = 0.92), while SSC exhibited strong associations with g_s_ (*r* = 0.92) and Tr (*r* = 0.94) (all *p* < 0.01).(3)Cross-module connectivity: Significant positive associations linked Chla to BDI (*r* = 0.90), LDW (*r* = 0.92), and TDW (*r* = 0.91); Chlb to LDW (r = 0.92); and TChl to TDW (*r* = 0.90). Similarly, SSC correlated positively with SDW (*r* = 0.90) and TDW (*r* = 0.91) (all *p* < 0.01). Notably, C_i_ exhibited negative correlations with all measured indicators (*r* = −0.740 to −0.229), which are significant for Chla, Chlb, and TChl (*p* < 0.05), and highly significant for Tr and SSC (*p* < 0.001). Collectively, this network topology underscores tight functional coupling among carbon assimilation, biomass partitioning, and nutrient metabolism.

#### 2.6.2. Comprehensive Evaluation of Fertilization Effects Based on Principal Component Analysis

A principal component analysis (PCA) was performed for all 15 traits across all fertilization treatments. The first two principal components together explained 88.31% of the total variance (Figure 7), effectively capturing most of the phenotypic variation. These two components were, thus, retained for use as composite variables, with the first principal component (PC1) exhibiting high positive loadings (>0.7) for all metrics except SSC (Figure 8). The low PC1 loading of SSC indicates its distinct regulatory pattern as a mobile carbon metabolite, whose accumulation responds rapidly to nutrient shifts independent of structural biomass traits. Based on the PCA results, a comprehensive evaluation model was derived as follows:*Y* = 0.816 *Y*_1_ + 0.067 *Y*_2_
where *Y*_1_ and *Y*_2_ represent PC1 and PC2 scores, respectively. The model ranked treatments as T7 > T9 > T6 > T11 > T10 > T12 > T5 > T13 > T3 > T14 > T8 > T4 > T2 > T1 (Table 6). This result indicates that optimal performance occurred under T7 (medium N, high P, medium K) followed by T9 (medium N, low P, low K).

## 3. Discussion

Primary morphological development—manifested through height and basal diameter increments—obeys stoichiometric scaling laws where phosphorus availability dictates structural investment thresholds in leguminous seedlings [36]. This study demonstrated that NPK fertilization significantly regulated morphological development (height and basal diameter increments) in *Dalbergia odorifera* T. C. Chen seedlings, with phosphorus (P) as the core regulator. Range analysis clearly revealed the fertilizer effect ordination as P > N > K, aligning with Zhao et al. [37] but contrasting with the conventional view of nitrogen (N) as the primary limiting factor [38,39,40]. This phosphorus dominance may arise from two mechanisms: (1) as a leguminous species, *D. odorifera* requires substantial P for root nodule development and nitrogen fixation, critical for energy metabolism and membrane biogenesis [41,42]; (2) the acidic red soil’s low native P availability was alleviated by supplementation, reducing aluminum/iron-induced P fixation [43,44]. Notably, both parameters exhibited single-peak responses to P, peaking at high P levels (T7 treatment), with 239.29% and 128.19% increases in height and diameter increments versus control. The exceptional performance of T7 arises from P-mediated synergy between enhanced nodule efficiency and photosynthetic capacity. This deviation from studies reporting peak responses at medium P [45] may reflect species-specific adaptation to P scarcity in tropical soils, where high investment in nodule function under abundant P maximizes growth returns. Differential analysis suggests that legume root nodules may compensate for soil N limitations, amplifying P’s direct regulation of stem elongation [41]. Collectively, P constitutes the primary limiting factor for *D. odorifera* seedling morphogenesis in P-deficient acidic soils.

Beyond morphological traits, fertilization critically shaped seedling biomass allocation patterns [46]. While the influence of nitrogen (N), phosphorus (P), and potassium (K) varies [47], leaf biomass typically dominates in most tree species under nutrient addition, with stem or root dominance being comparatively rare [48,49]. In contrast, this study demonstrated that *D. odorifera* seedlings exhibit a distinct stem-biased allocation strategy under fertilization (stem:52% > root:26% > leaf:22%). This pronounced stem allocation aligns with *Torreya grandis* Fortune ex Lindl [50] but contrasts sharply with the leaf-dominant pattern in species like *Cunninghamia lanceolata* (Lamb.) Hook. and *Camphora micrantha* (Hayata) Y. Yang, Bing Liu & Zhi Yang [51,52]. Univariate analysis revealed continuous increases in stem dry weight with P supply (peaking at high P), whereas leaf dry weight showed a single-peak response (maximum at medium P). This divergence reflects functional prioritization: SDW benefits from sustained carbon allocation to structural tissues, while LDW declines post saturation due to resource reallocation toward stem storage. The unexpected decline in leaf biomass at high P levels may indicate a strategic shift toward structural investment once photosynthetic capacity is saturated, prioritizing stem development for resource storage. This suggests P drives stem dominance through dual mechanisms: (1) directly by activating sucrose phosphate synthase to enhance photosynthetic carbon conversion into cellulose, accelerating xylem development; (2) indirectly by inducing root morphological adaptations that improve nutrient acquisition, thereby supporting stem accumulation [53]. Notably, the fertilizer effect ordination for stem dry weight (P > K > N) diverged from the N-dominated pattern in non-nitrogen-fixing species like *Eucalyptus urophylla × grandis* [54,55]. This adaptive strategy likely reflects an evolutionary response to tropical environments, prioritizing stem development for mechanical support and establishing the foundation for heartwood formation.

Concomitant with structural changes, precision fertilization significantly enhanced photosynthetic performance in *D. odorifera* seedlings, with phosphorus (P) identified as the key regulatory factor [37]. The high-phosphorus treatment (T7) increased chlorophyll a, b, and total chlorophyll content by 68–75% versus control, with P showing the highest regulatory weight among elements (P > N > K). This contrasts with studies reporting chlorophyll b peaking at moderate P levels in other species [53], suggesting species-specific P utilization strategies. We hypothesize that optimal chlorophyll synthesis at high P stems from P’s dual role: as an ATP energy carrier and metabolic regulator, it facilitates Mg^2+^ transport to chloroplasts, enabling efficient porphyrin ring synthesis [25]. Correspondingly, P_n_ increased by 82% under T7 versus control, with the significant negative P_n_-C_i_ correlation indicating non-stomatal limitation dominance. This aligns with the P-mediated activation of Calvin cycle enzymes (e.g., Rubisco) and photophosphorylation [56]. Furthermore, P exerted hierarchical control over carbon–nitrogen metabolites. Soluble sugar content peaked under T7 (+139%, P-regulated) and correlated strongly with g_s_, confirming that P-enhanced carbon assimilation drives sugar accumulation. Conversely, soluble protein content peaked under T9 (+226%, K-regulated), reflecting P-K synergy in nitrogen metabolism. Although P participates in protein synthesis, maximal accumulation depended on high potassium (K > N > P). This K-dominance aligns with the legume nitrogen fixation strategy, as reduced soil N dependence amplifies K’s role in osmoregulation and enzyme activation [57]. Crucially, the strong SDW–P_n_ correlation revealed photosynthetic carbon assimilation as the primary driver of stem accumulation. P may further mediate sucrose allocation to stems via sucrose transporters, providing metabolic support for biomass partitioning.

Integrating these physiological responses, principal component analysis (PCA) identified the T7 treatment (N2P3K2) as the optimal formula (cumulative contribution rate: 88.312%). Its core advantage lies in the synergistic mechanism of phosphorus-mediated photosynthetic carbon assimilation and stem biomass translocation. Building on this finding, we propose practical implications for cultivation and conservation: the T7 formula is recommended for nursery production to maximize growth in acidic red soils, where high-P fertilization counteracts aluminum/iron fixation while supporting nodule development. For wild population conservation, targeted P supplementation in degraded habitats could enhance seedling establishment. Future domestication programs should prioritize the selection of P-efficient genotypes to reduce fertilizer dependency, though caution is warranted against excessive K application (e.g., T9), which may disrupt C:N balance and compromise long-term stress resilience despite boosting soluble proteins [41].

Methodologically, this study advances beyond previous morphology-focused approaches [31,32,33] by quantifying the coupling between stem biomass allocation and soluble sugar content, establishing a theoretical basis for seedling standards through source–sink regulation. However, three limitations require acknowledgment: (1) unresolved phosphorus signaling; the regulatory mechanism of PHR transcription factors on sucrose transporter genes remains unelucidated; (2) unassessed soil–root interactions; rhizosphere P dynamics and mycorrhizal colonization post fertilization were unmonitored; and (3) insufficient maturation linkage; connections between P supply, stem accumulation, and heartwood formation remain unmodelled. To address these gaps, future research should (1) apply multi-omics to decipher P signaling pathways in *D. odorifera* growth; (2) integrate rhizosphere microbiome analysis to optimize P fertilizer strategies; and (3) extend observations to establish fertilization–heartwood accumulation correlations.

## 4. Materials and Methods

### 4.1. Site Area

This study was conducted within the experimental nursery of the Hainan Academy of Forestry (Hainan Academy of Mangrove), located in Yunlong Town, Haikou City, Hainan Province, China (19°52′ N, 110°28′ E). This region experiences a tropical maritime climate and warm temperatures (mean annual: 24.9 °C), with summer maxima exceeding 30 °C and winter minima averaging 20 °C (Figure 9). Annual precipitation ranges from 1500 to 2000 mm and predominantly occurs during the summer-to-autumn monsoon season. The study site receives >2000 annual sunshine hours with continuously high solar irradiance, as is characteristic of coastal tropical environments [58].

### 4.2. Materials

Three-year-old *Dalbergia odorifera* T. C. Chen seedlings with uniform growth (height: 117.11 ± 15.83 cm; basal diameter: 16.57 ± 2.46 mm) were transplanted into non-woven fabric nursery bags (30 cm × 30 cm) in March 2023. The growth substrate consisted of red soil, peat, and perlite (2:1:1, *v*/*v*/*v*) with the following initial properties: pH 4.9, total N 6.82 g/kg, total P 0.69 g/kg, total K 3.42 g/kg. The fertilizers used in this experiment were urea (N, 46%), calcium superphosphate (P_2_O_5_, 12%), and potassium sulfate (K_2_O, 60%).

### 4.3. Experimental Design

A “3414” incomplete orthogonal regression design was implemented with three factors (N, P, K) at four levels: Level 0 (control, no fertilizer), Level 1 (low, 50% conventional rate), Level 2 (moderate, 100% conventional rate) and Level 3 (high, 150% conventional rate) [28]. Conventional rates were urea 4.31 g/plant, superphosphate 20.83 g/plan, and potassium sulfate 0.83 g/plant [31,32,33]. Each treatment was arranged in a randomized block design with three biological replicates, each containing three *D. odorifera* seedlings (*n* = 126 total) (Table 7). The trial lasted 10 months (March–December 2023). Phosphorus was applied as basal fertilizer in March, while nitrogen fertilizer and potassium fertilizer were divided into three equal parts and applied through a drip irrigation system in three split applications (in April, July, and October, respectively). Irrigation was standardized via drip systems to maintain soil moisture at 75–80% field capacity.

### 4.4. Measurement Parameters

(1)Growth and biomass parameters

Four randomly selected seedlings were sampled per treatment at experiment initiation (March 2023) and termination (December 2023). The seedling height (SH; cm) and basal diameter (BD; mm; measured at the ground) were measured using steel tape (8208, Deli Group Co., Ltd., Ningbo, China) and digital calipers (SHG-200MM, Shanghai Tool Works Co., Ltd., Shanghai, China), respectively. Growth over the experimental period was calculated by subtracting the initial measurements from the final values. After thoroughly rinsing the harvested seedlings, the leaves, stems, and roots were separated. Tissues were oven-dried (105 °C for 30 min to ensure enzyme deactivation, then 80 °C to constant mass), and the dry weight of the leaves (LDW; g), stems (SDW; g), and roots (RDW; g) was recorded using an analytical balance, as well as the total weight (TDW; g) [59].

(2)Leaf physiological parameters

Fifteen days prior to experimental termination, four representative seedlings exhibiting consistent growth were selected per treatment. Mature functional leaves (from third to fourth position below the terminal bud of new shoots) were excised, immediately flash-frozen in liquid nitrogen, and transported to the laboratory for analysis. Soluble sugar, soluble protein, and chlorophyll content were quantified as key indicators of plant nutritional status and growth vigor, given their respective roles: soluble sugar as the core substrate for energy metabolism, soluble protein in enzymatic catalysis and stress responses, and chlorophyll as a direct proxy for photosynthetic capacity [60].

Soluble sugar content (SSC) was determined via the anthrone colorimetric method [61]. Frozen leaf tissue (0.2 g) was pulverized, homogenized in 5 mL distilled water, and extracted in a boiling water bath for 30 min. After cooling, the homogenate was centrifuged (4000× *g*, 10 min). The supernatant was adjusted to 25 mL with distilled water. An aliquot (1 mL) was reacted with 5 mL anthrone reagent, boiled for 10 min, cooled, and absorbance measured at 620 nm using a spectrophotometer (UV-1050, Shanghai Techcomp Scientific Instrument Co., Ltd., Shanghai, China). Glucose standards were used to generate the calibration curve for quantification.

Soluble protein content (SPC) was assayed using Coomassie Brilliant Blue G-250 (Shanghai Macklin Biochemical Co., Ltd., Shanghai, China) [61]. Frozen leaves (0.2 g) were homogenized in 5 mL phosphate buffer (pH 7.0) and centrifuged (4000× *g*, 10 min). The supernatant (0.1 mL) was mixed with 5 mL Coomassie Brilliant Blue reagent, incubated for 5 min, and absorbance read at 595 nm. Quantification was performed using a bovine serum albumin standard curve.

Chlorophyll content was determined by propanol extraction [60]. Fresh leaves (0.5 g, main vein excised) were finely chopped and homogenized with 5 mL propanol (analytical grade) and quartz sand under ice-cold conditions. The homogenate was transferred to a centrifuge tube, diluted to 10 mL with propanol, and incubated at 4 °C in darkness for 24 h with intermittent agitation. Following centrifugation (4000× *g*, 10 min), supernatant absorbance was measured at 663 nm (chlorophyll a) and 645 nm (chlorophyll b) against a propanol blank using a spectrophotometer (UV-1050, Shanghai Techcomp Scientific Instrument Co., Ltd., Shanghai, China). Chlorophyll concentrations were calculated according to Arnon’s equations:Chlorophyll a (mg/g) = [(12.7 × A_663_) − (2.69 × A_645_)] × V/(1000 × W)Chlorophyll b (mg/g) = [(22.9 × A_645_) − (4.68 × A_663_)] × V/(1000 × W)Total Chlorophyll = Chlorophyll a + Chlorophyll b
where A denotes absorbance, V is extract volume (mL), and W is sample fresh weight (g).

(3)Photosynthetic characterization

Leaf photosynthetic parameters were measured between 09:00 and 11:00 on sunny days during mid-October 2023 using an LI-6800 portable photosynthesis system (LI-COR Biosciences, Lincoln, NE, USA). For each seedling, three mature functional leaves from the upper canopy were selected for measurement. The following settings were used: 500 μmol/s airflow, 400 μmol/mol reference CO_2_ level, 25 °C leaf chamber temperature, 60% relative humidity, and 10,000 rpm cooling fan speed. Recorded parameters included net photosynthetic rate (P_n_), stomatal conductance (g_s_), transpiration rate (T_r_), and intercellular CO_2_ concentration (C_i_). The mean value of three leaves represented the measurement for each seedling [61].

### 4.5. Data Analysis

The experimental data were analyzed using Microsoft Excel (v.2019). Analysis of variance (ANOVA) was employed to statistically assess the overall impact of 15 key parameters encompassing growth indices (height increment, basal diameter increment, and dry weights of various organs), leaf physiological parameters (e.g., soluble sugar, soluble protein), and photosynthetic parameters (e.g., net photosynthetic rate, stomatal conductance). The original data were log-transformed to satisfy the normality and homoscedasticity assumptions of ANOVA. The means of 15 key parameters were compared via Duncan’s new multiple range method, and ith statistical significance was defined at *p* < 0.05. All statistical analyses were conducted using SPSS Statistics (v.26.0). Optimal fertilizer ratios were comprehensively evaluated using a principal component analysis (PCA). Results were visualized in OriginPro (v.2021b) with standard formatting; error bars represent standard errors (SEs). All tabular and graphical data presentations illustrate the mean ± SE.

## 5. Conclusions

This study identified phosphorus (P) as the critical regulatory factor governing *Dalbergia odorifera* T. C. Chen seedling development in acidic phosphorus-fixing soils. The high-phosphorus formulation (N2P3K2) demonstrated optimal performance by coordinately enhancing three physiological domains: morphological development through accelerated structural growth, stem-dominant biomass partitioning via carbon reallocation mechanisms, and photosynthetic improvement through biochemical optimization. Crucially, metabolic coordination was evidenced by the interdependent accumulation of carbon metabolites and nitrogenous compounds. Principal component analysis confirmed the treatment’s superiority, revealing P’s integrative role in synchronizing carbon assimilation with stem translocation. This mechanistic insight establishes a physiological framework linking stem biomass accumulation with carbohydrate dynamics for quality seedling production. Future validation should quantify mycorrhizal contributions to P mobilization, maturation-phase heartwood formation under sustained P supply, and the transcriptomic regulation of K-dependent protein synthesis, advancing the precision conservation of this endangered species.

## Figures and Tables

**Figure 1 plants-14-02545-f001:**
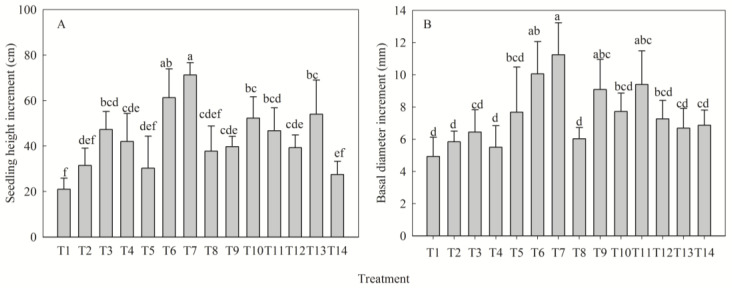
Effects of different fertilization treatments on height increment (**A**) and basal diameter increment (**B**) of *Dalbergia odorifera* T. C. Chen seedlings. Fertilization treatments: T1(N0P0K0), T2(N0P2K2), T3(N1P2K2), T4(N2P0K2), T5(N2P1K2), T6(N2P2K2), T7(N2P3K2), T8(N2P2K0), T9(N2P2K1), T10(N2P2K3), T11(N3P2K2), T12(N1P1K2), T13(N1P2K1), T14(N2P1K1). Data are presented as mean ± SE (*n* = 3 biological replicates, 3 seedlings per replicate). Different lowercase letters within each subplot indicate significant differences (*p* < 0.05) determined by one-way ANOVA with Duncan’s post hoc test.

**Figure 2 plants-14-02545-f002:**
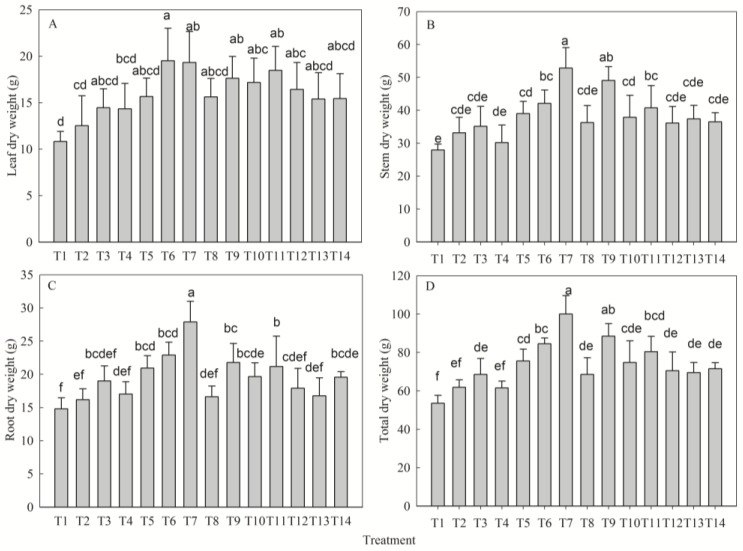
Effect of different fertilization treatments on leaf dry weight (**A**), stem dry weight (**B**), root dry weight (**C**), and total dry weight (**D**) of *D. odorifera* seedlings. Fertilization treatments: T1(N0P0K0), T2(N0P2K2), T3(N1P2K2), T4(N2P0K2), T5(N2P1K2), T6(N2P2K2), T7(N2P3K2), T8(N2P2K0), T9(N2P2K1), T10(N2P2K3), T11(N3P2K2), T12(N1P1K2), T13(N1P2K1), T14(N2P1K1). Data are presented as mean ± SE (*n* = 3 biological replicates, 3 seedlings per replicate). Different lowercase letters within each subplot indicate significant differences (*p* < 0.05) determined by one-way ANOVA with Duncan’s post hoc test.

**Figure 3 plants-14-02545-f003:**
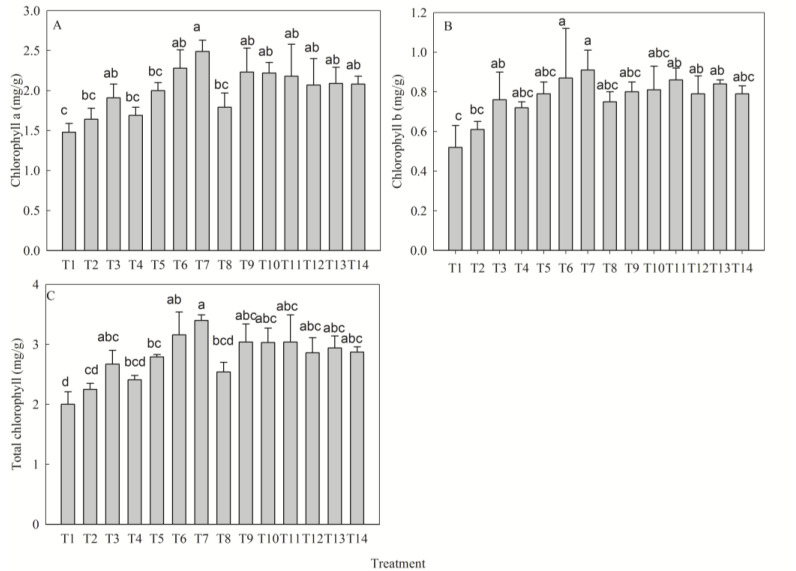
Effect of different fertilization treatments on chlorophyll a (**A**), chlorophyll b (**B**), and total chlorophyll (**C**) of *D. odorifera* seedlings. Fertilization treatments: T1(N0P0K0), T2(N0P2K2), T3(N1P2K2), T4(N2P0K2), T5(N2P1K2), T6(N2P2K2), T7(N2P3K2), T8(N2P2K0), T9(N2P2K1), T10(N2P2K3), T11(N3P2K2), T12(N1P1K2), T13(N1P2K1), T14(N2P1K1). Data are presented as mean ± SE (*n* = 3 biological replicates, 3 seedlings per replicate). Different lowercase letters within each subplot indicate significant differences (*p* < 0.05) determined by one-way ANOVA with Duncan’s post hoc test.

**Figure 4 plants-14-02545-f004:**
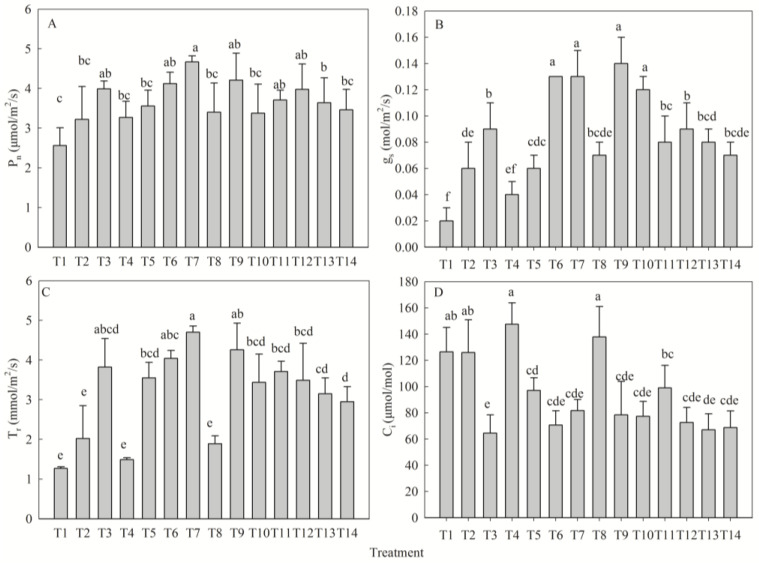
Effects of different fertilization treatments on net photosynthetic rate (**A**), stomatal conductance (**B**), transpiration rate (**C**), and intercellular CO_2_ concentration (**D**) of *D. odorifera* seedlings. Analysis indices: net photosynthetic rate (P_n_), stomatal conductance (g_s_), transpiration rate (T_r_), intercellular CO_2_ concentration (C_i_). Fertilization treatments: T1(N0P0K0), T2(N0P2K2), T3(N1P2K2), T4(N2P0K2), T5(N2P1K2), T6(N2P2K2), T7(N2P3K2), T8(N2P2K0), T9(N2P2K1), T10(N2P2K3), T11(N3P2K2), T12(N1P1K2), T13(N1P2K1), T14(N2P1K1). Data are presented as mean ± SE (*n* = 3 biological replicates, 3 seedlings per replicate). Different lowercase letters within each subplot indicate significant differences (*p* < 0.05) determined by one-way ANOVA with Duncan’s post hoc test.

**Figure 5 plants-14-02545-f005:**
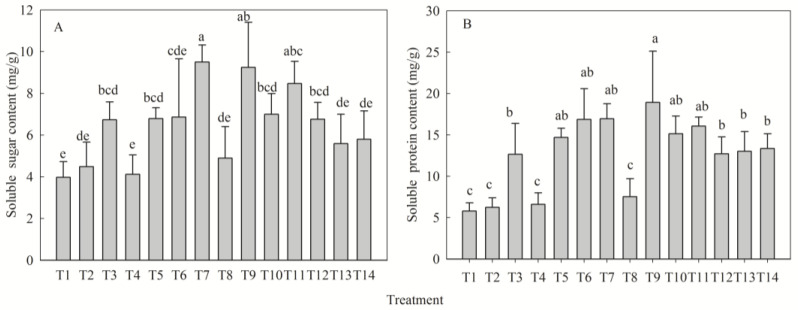
Effect of different fertilization treatments on soluble sugar (**A**) and soluble protein (**B**) content of *D. odorifera* seedlings. Fertilization treatments: T1(N0P0K0), T2(N0P2K2), T3(N1P2K2), T4(N2P0K2), T5(N2P1K2), T6(N2P2K2), T7(N2P3K2), T8(N2P2K0), T9(N2P2K1), T10(N2P2K3), T11(N3P2K2), T12(N1P1K2), T13(N1P2K1), T14(N2P1K1). Data are presented as mean ± SE (*n* = 3 biological replicates, 3 seedlings per replicate). Different lowercase letters within each subplot indicate significant differences (*p* < 0.05) determined by one-way ANOVA with Duncan’s post hoc test.

**Figure 6 plants-14-02545-f006:**
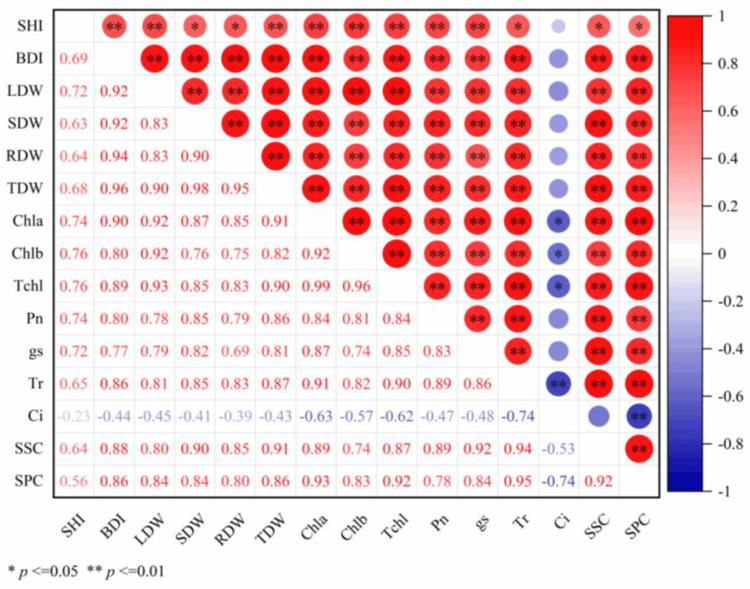
Heatmap depicting correlations among 15 parameters of *D. odorifera* seedlings across different fertilization treatments. * indicates significant correlation (*p* < 0.05), ** indicates extremely significant correlation (*p* < 0.01). Analysis indices: SHI (seedling height increment), BDI (basal diameter increment), LDW (leaf dry weight), SDW (stem dry weight), RDW (root dry weight), TDW (total dry weight), Chla (chlorophyll a), Chlb (chlorophyll b), Tchl (total chlorophyll), P_n_ (net photosynthetic rate), g_s_ (stomatal conductance), T_r_ (transpiration rate), C_i_ (intercellular CO_2_ concentration), SSC (soluble sugar content), SPC (soluble protein content).

**Figure 7 plants-14-02545-f007:**
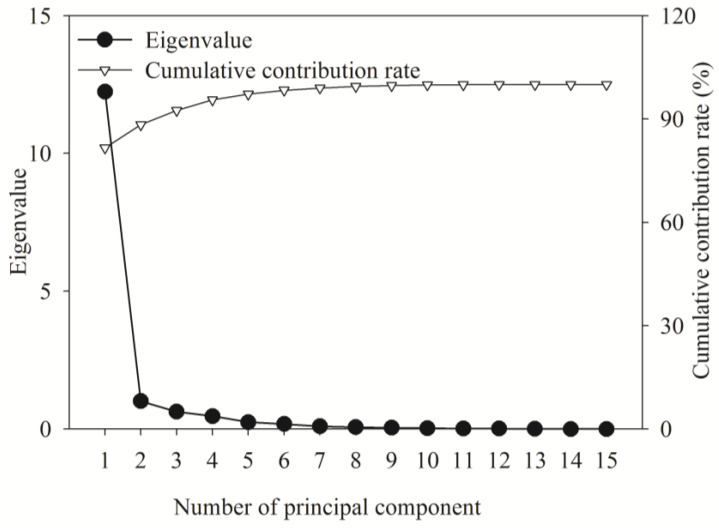
Principal component analysis of variance explained by fertilization on the growth of *D. odorifera* seedlings.

**Figure 8 plants-14-02545-f008:**
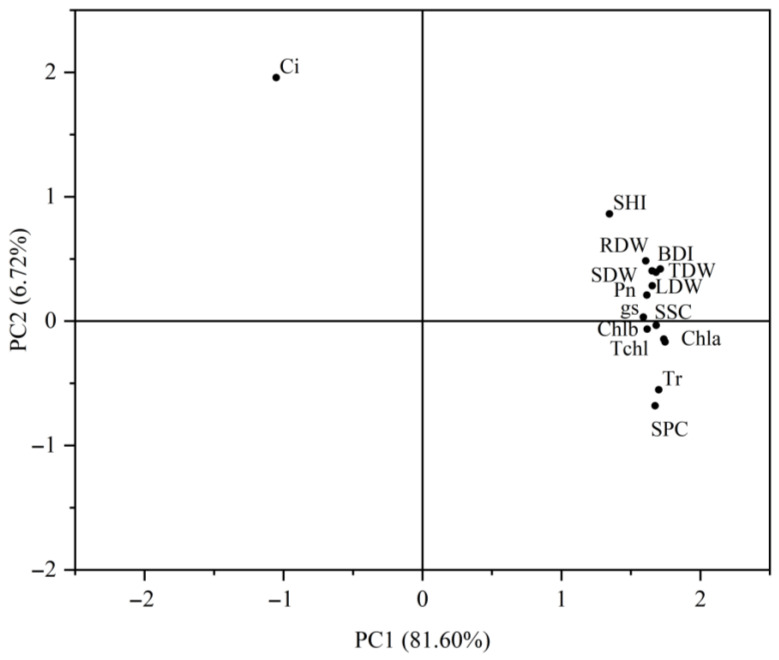
Component plots of principal component 1 and principal component 2. Analysis indices: SHI (seedling height increment), BDI (basal diameter increment), LDW (leaf dry weight), SDW (stem dry weight), RDW (root dry weight), TDW (total dry weight), Chla (chlorophyll a), Chlb (chlorophyll b), Tchl (total chlorophyll), P_n_ (net photosynthetic rate), g_s_ (stomatal conductance), T_r_ (transpiration rate), C_i_ (intercellular CO_2_ concentration), SSC (soluble sugar content), SPC (soluble protein content).

**Figure 9 plants-14-02545-f009:**
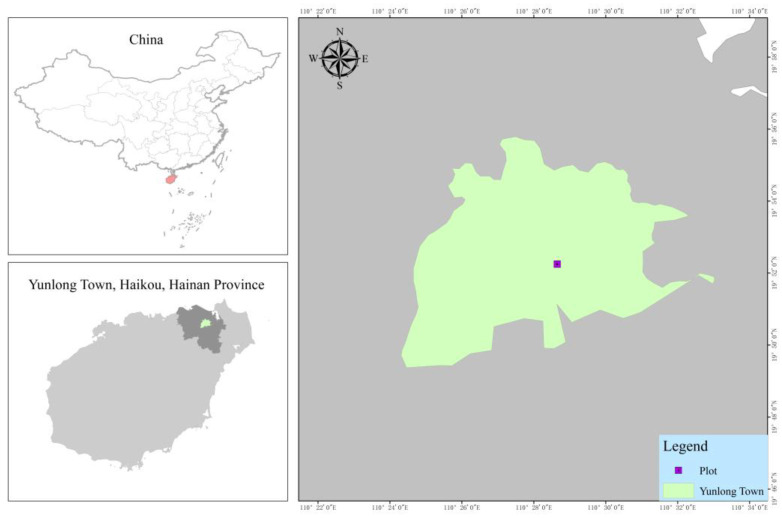
Schematic diagram of the geographical location of the study area.

**Table 1 plants-14-02545-t001:** Range analysis of the effects of different fertilization treatments on seedling morphogenesis. Range analysis was performed to quantify fertilizer effects using the range difference method, R_j_ = max(X_j_) − min(X_j_), where R_j_ is the range value of nutrient j (N, P, or K) and X_j_ is the mean trait response at each nutrient level.

Parameter	Factor	Level 0	Level 1	Level 2	Level 3	Range Value
Seedling height increment (cm)	N	31.50 ± 7.57 b	47.25 ± 7.95 ab	61.25 ± 12.68 a	46.75 ± 10.03 ab	29.75
	P	42.00 ± 12.35 bc	30.25 ± 5.40 c	61.25 ± 12.68 ab	71.25 ± 14.10 a	41.00
	K	37.75 ± 11.03 b	39.75 ± 4.55 b	61.25 ± 12.68 a	52.25 ± 9.42 ab	23.50
Basal diameter increment (mm)	N	5.85 ± 0.66 b	6.45 ± 1.39 b	10.06 ± 2.09 a	9.40 ± 2.00 a	4.21
	P	5.51 ± 1.34 b	7.67 ± 1.98 ab	10.06 ± 2.09 a	11.25 ± 2.81 a	5.74
	K	6.03 ± 0.69 b	9.09 ± 1.86 a	10.06 ± 2.09 a	7.72 ± 1.14 ab	3.37

Different lowercase letters within each subplot indicate significant differences (*p* < 0.05) determined by one-way ANOVA with Duncan’s post hoc test.

**Table 2 plants-14-02545-t002:** Range analysis of the effect of different fertilization treatments on seedling biomass. Range analysis was performed to quantify fertilizer effects using the range difference method, R_j_ = max(X_j_) − min(X_j_), where R_j_ is the range value of nutrient j (N, P, or K) and X_j_ is the mean trait response at each nutrient level.

Parameter	Factor	Level 0	Level 1	Level 2	Level 3	Range Value
Leaf dry weight (g)	N	12.53 ± 3.23 b	14.46 ± 2.04 ab	19.51 ± 3.50 a	18.48 ± 2.59 ab	6.98
	P	14.34 ± 2.74 b	15.66 ± 1.97 b	19.51 ± 3.49 a	19.33 ± 3.32 a	5.17
	K	15.63 ± 1.98 a	17.62 ± 2.36 a	19.51 ± 3.49 a	17.18 ± 2.61 a	3.88
Stem dry weight (g)	N	33.16 ± 4.70 b	35.14 ± 6.08 ab	42.11 ± 4.05 a	40.75 ± 6.77 a	8.95
	P	30.21 ± 5.31 c	38.99 ± 3.72 bc	42.11 ± 4.05 b	52.82 ± 6.25 a	22.61
	K	36.29 ± 5.18 b	49.06 ± 4.21 a	42.11 ± 4.05 a	37.89 ± 6.66 b	12.77
Root dry weight (g)	N	16.18 ± 1.65 b	19.01 ± 2.26 ab	22.9 ± 1.94 a	21.18 ± 4.57 ab	6.72
	P	17.00 ± 1.89 b	20.94 ± 1.87 b	22.9 ± 1.94 a	27.88 ± 3.12 a	10.88
	K	16.63 ± 1.62 c	21.79 ± 2.86 ab	22.9 ± 1.94 a	19.66 ± 2.07 bc	6.28
Total dry weight (g)	N	61.87 ± 3.87 b	68.61 ± 8.26 b	84.52 ± 3.00 a	80.40 ± 8.04 a	22.65
	P	61.56 ± 3.52 c	75.59 ± 6.15 b	84.52 ± 3.00 ab	100.03 ± 9.59 a	38.47
	K	68.54 ± 8.77 b	88.47 ± 6.61 a	84.52 ± 3.00 a	74.73 ± 11.32 ab	19.93

Different lowercase letters within each subplot indicate significant differences (*p* < 0.05) determined by one-way ANOVA with Duncan’s post hoc test.

**Table 3 plants-14-02545-t003:** Range analysis of the effect of different fertilization treatments on photosynthetic pigment. Range analysis was performed to quantify fertilizer effects using the range difference method, R_j_ = max(X_j_) − min(X_j_), where R_j_ is the range value of nutrient j (N, P, or K) and X_j_ is the mean trait response at each nutrient level.

Parameter	Factor	Level 0	Level 1	Level 2	Level 3	Range Value
Chlorophyll a (mg/g)	N	1.64 ± 0.14 b	1.91 ± 0.17 ab	2.28 ± 0.23 a	2.18 ± 0.40 a	0.64
	P	1.69 ± 0.10 c	2.00 ± 0.10 b	2.28 ± 0.23 ab	2.49 ± 0.14 a	0.80
	K	1.79 ± 0.18 b	2.23 ± 0.30 a	2.28 ± 0.23 a	2.22 ± 0.13 a	0.49
Chlorophyll b (mg/g)	N	0.61 ± 0.04 a	0.76 ± 0.14 a	0.87 ± 0.25 a	0.86 ± 0.06 a	0.26
	P	0.72 ± 0.03 a	0.79 ± 0.06 a	0.87 ± 0.25 a	0.91 ± 0.10 a	0.19
	K	0.75 ± 0.05 a	0.80 ± 0.05 a	0.87 ± 0.25 a	0.81 ± 0.12 a	0.12
Total chlorophyll (mg/g)	N	2.25 ± 0.10 b	2.67 ± 0.23 ab	3.16 ± 0.38 a	3.04 ± 0.45 a	0.91
	P	2.41 ± 0.07 c	2.79 ± 0.04 b	3.16 ± 0.38 ab	3.40 ± 0.09 a	0.99
	K	2.54 ± 0.16 b	3.04 ± 0.30 ab	3.16 ± 0.38 a	3.03 ± 0.24 ab	0.62

Different lowercase letters within each subplot indicate significant differences (*p* < 0.05) determined by one-way ANOVA with Duncan’s post hoc test.

**Table 4 plants-14-02545-t004:** Range analysis of the effects of different fertilization treatments on gas exchange parameters. Range analysis was performed to quantify fertilizer effects using the range difference method, R_j_ = max(X_j_) − min(X_j_), where R_j_ is the range value of nutrient j (N, P, or K) and X_j_ is the mean trait response at each nutrient level.

Parameter	Factor	Level 0	Level 1	Level 2	Level 3	Range Value
Net photosynthetic rate (μmol/m^2^/s)	N	3.22 ± 0.83 b	3.99 ± 0.20 ab	4.12 ± 0.29 a	3.71 ± 0.25 ab	0.9
	P	3.27 ± 0.41 c	3.56 ± 0.40 bc	4.12 ± 0.29 ab	4.67 ± 0.15 a	1.4
	K	3.40 ± 0.74 a	4.21 ± 0.68 a	4.12 ± 0.29 a	3.38 ± 0.73 a	0.81
Stomatal conductance (mol/m^2^/s)	N	0.06 ± 0.02 bc	0.09 ± 0.02 b	0.13 ± 0.02 a	0.08 ± 0.02 c	0.06
	P	0.04 ± 0.01 b	0.06 ± 0.01 b	0.13 ± 0.02 a	0.14 ± 0.02 a	0.09
	K	0.07 ± 0.01 b	0.14 ± 0.02 a	0.13 ± 0.02 a	0.12 ± 0.01 a	0.07
Transpiration rate (mol/m^2^/s)	N	2.02 ± 0.83 b	3.82 ± 0.72 a	4.04 ± 0.20 a	3.71 ± 0.26 a	2.02
	P	1.49 ± 0.05 d	3.55 ± 0.39 c	4.04 ± 0.20 b	4.70 ± 0.16 a	3.21
	K	1.89 ± 0.20 b	4.26 ± 0.67 a	4.04 ± 0.20 a	3.44 ± 0.71 a	2.37
Intercellular CO_2_ concentration (μmol/m^2^/s)	N	125.95 ± 24.88 a	64.45 ± 13.87 c	70.63 ± 10.89 bc	98.98 ± 17.08 ab	61.50
	P	147.55 ± 16.31 a	97.05 ± 9.61 b	70.63 ± 10.89 bc	81.73 ± 8.35 b	76.93
	K	137.80 ± 23.30 a	78.40 ± 25.48 a	70.63 ± 10.89 bc	77.28 ± 11.43 c	60.52

Different lowercase letters within each subplot indicate significant differences (*p* < 0.05) determined by one-way ANOVA with Duncan’s post hoc test.

**Table 5 plants-14-02545-t005:** Range analysis of the effect of different fertilization treatments on soluble sugar and soluble protein content. Range analysis was performed to quantify fertilizer effects using the range difference method, R_j_ = max(X_j_) − min(X_j_), where R_j_ is the range value of nutrient j (N, P, or K) and X_j_ is the mean trait response at each nutrient level.

Parameter	Factor	Level 0	Level 1	Level 2	Level 3	Range Value
Soluble sugar content (mg/g)	N	4.49 ± 1.18 b	6.73 ± 0.86 ab	6.86 ± 2.79 ab	8.46 ± 2.79 a	3.97
	P	4.12 ± 0.93 b	6.79 ± 0.52 ab	6.86 ± 2.79 ab	9.50 ± 0.82 a	5.38
	K	4.90 ± 1.50 b	9.24 ± 2.16 a	6.86 ± 2.79 ab	6.99 ± 1.00 ab	4.34
Soluble protein content (mg/g)	N	6.25 ± 1.13 b	12.66 ± 3.72 ab	16.86 ± 3.72 a	16.06 ± 1.09 a	10.61
	P	6.62 ± 1.38 b	14.69 ± 1.12 a	16.86 ± 3.72 a	16.94 ± 1.81 a	10.32
	K	7.53 ± 2.19 b	18.94 ± 6.18 a	16.86 ± 3.72 a	15.13 ± 2.16 a	11.41

Different lowercase letters within each subplot indicate significant differences (*p* < 0.05) determined by one-way ANOVA with Duncan’s post hoc test.

**Table 6 plants-14-02545-t006:** Comprehensive scores and rankings of the main components of each fertilization treatment.

Fertilization Treatment	1st Principal Component Score (*Y*_1_)	2nd Principal Component Score (*Y*_1_)	Aggregate Score (*Y*)	Rank
T1	−6.685580906	0.025042903	−5.093657638	14
T2	−4.214583572	0.587257653	−3.155666934	13
T3	−0.117797928	−1.063754402	−0.192858634	9
T4	−3.935690246	1.490184221	−2.855595941	12
T5	0.169408269	−0.759485148	0.055544937	7
T6	3.355158235	0.612888541	2.616852135	3
T7	6.415965173	1.801764641	5.065142429	1
T8	−2.529796481	1.281857014	−1.804138241	11
T9	3.667974009	−0.31744776	2.765155675	2
T10	1.368564003	−0.470855918	0.997566681	5
T11	2.574127645	−0.135080961	1.949040803	4
T12	0.330418012	−0.907324911	0.163950419	6
T13	0.061868	−0.861239638	−0.036286625	8
T14	−0.460048996	−1.283797544	−0.475059492	10

**Table 7 plants-14-02545-t007:** Fourteen different experimental fertilizer treatments based on the “3414” optimal design scheme.

Number	Fertilization Treatment	N (g/plant)	P_2_O_5_ (g/plant)	K_2_O (g/plant)
T1	N0P0K0	0.00	0.00	0.00
T2	N0P2K2	0.00	20.83	0.83
T3	N1P2K2	2.16	20.83	0.83
T4	N2P0K2	4.31	0.00	0.83
T5	N2P1K2	4.31	10.42	0.83
T6	N2P2K2	4.31	20.83	0.83
T7	N2P3K2	4.31	31.25	0.83
T8	N2P2K0	4.31	20.83	0.00
T9	N2P2K1	4.31	20.83	0.42
T10	N2P2K3	4.31	20.83	1.25
T11	N3P2K2	6.47	20.83	0.83
T12	N1P1K2	2.16	10.42	0.83
T13	N1P2K1	2.16	20.83	0.42
T14	N2P1K1	2.16	10.42	0.42

## Data Availability

Data are contained within the article.

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
