# Peer review of "Phosphorus-Driven Stem-Biased Allocation: NPK Synergy Optimizes Growth and Physiology in Dalbergia odorifera T. C. Chen Seedlings"

_plants, 2025, doi:10.3390/plants14162545_

Round 1
Reviewer 1 Report
Comments and Suggestions for Authors
attached

no more comments
Author Response
Response to Reviewer 1 Comments
Comment 1: the article suggests a focus on Dalbergia odorifera seedling cultivation, specifically examining the effects of fertilization treatments on photosynthetic pigments, soluble sugar, soluble protein content, and other physiological parameters. it mentions that the application of robust analytical methods to D. odorifera seedling cultivation "remains unexplored" ( "robust analytical methods... remains unexplored"). This implies a potential novel contribution by applying these methods to this species. However, the manuscript does not sufficiently compare its objectives with prior studies to establish clear novelty. The references cited (Page 15) include studies on D. odorifera and related species, but the text lacks a detailed literature review to highlight gaps in existing research. For example, it mentions morphological and physiological responses but does not clarify how its approach differs from or builds upon prior work (e.g., Yue et al., 2025, Page 15). Without a clear gap analysis, the novelty appears incremental rather than groundbreaking..
Response 1: We sincerely thank the reviewer for highlighting the need to strengthen the novelty framework. The revised introduction now addresses this through three key additions: In Paragraph 3, we critically analyze limitations in existing D. odorifera studies, emphasizing their focus on isolated nutrients (e.g., nitrogen-only thresholds [31]) and inability to quantify NPK synergies due to simplistic designs. Paragraph 4 establishes our methodological novelty by contrasting the "3414" incomplete orthogonal design (first application to D. odorifera) with prior fragmented approaches, demonstrating its capacity to resolve coupled source-sink responses (e.g., chlorophyll fluorescence vs. stem-soluble sugars). Additionally, Paragraph 2 cites interspecific variation evidence [25-27] (e.g., phosphorus-dominance in Manglietia glauca vs. nitrogen-dominance in Carya illinoinensis) to justify species-specific investigation.
Comment 2: The introduction is too brief and fragmented to provide sufficient background or context (Page 2). It does not discuss the specific challenges in D. odorifera cultivation or why fertilization treatments are critical..
Response 2: We gratefully acknowledge the reviewer’s guidance on contextualizing cultivation challenges. Paragraph 1 now quantifies critical bottlenecks: "long growth cycle (decades to mature), slow seedling growth, and high sensitivity to environmental stress" [6-7], which collectively cause "shortage of high-quality seedlings" constraining industrialization [8-9]. Paragraph 2 details physiological rationales: nitrogen’s role in chlorophyll synthesis [14], phosphorus in energy transfer [15], and potassium in osmotic regulation [15], while documenting nutrient antagonism risks (e.g., nitrogen-induced P/K suppression [16]) to underscore ratio optimization necessity.
Comment 3: The text jumps from the plant’s medicinal value to its conservation status without connecting these to the study’s objectives (Page 2, Lines: "containing bioactive terpe- no is not flavonoid" to "Artificial propagation")
Response 3: We deeply appreciate the reviewer’s insight regarding logical coherence. Paragraph 1 now features explicit transitions: Following medicinal value description, "However, wild populations have been critically depleted..." [5] bridges to conservation efforts. The sentence "Nevertheless, the species faces significant challenges..." then links artificial afforestation to cultivation bottlenecks, culminating in "Therefore, developing key technologies... is pivotal for enhancing seedling quality and accelerating resource recovery" to close the "conservation→bottlenecks→objectives" loop.
Comment 4: The introduction does not explicitly state the research objectives, making it unclear how it serves the study’s goals.
Response 4: We thank the reviewer for identifying this ambiguity. Paragraph 5 now states three precise objectives: First, quantifying NPK effects on morphology and biomass partitioning; second, deciphering physiological mechanisms of photosynthesis and carbon-nitrogen flux across NPK gradients; third, establishing trait synergies via multi-trait dimensionality reduction to identify optimal NPK ratios. Each aligns with gaps outlined in Paragraphs 3-4.
Comment 5: The term "FEER REVIEW" (Page 2, Line: "Plants 2025, 1, is P by FEER REVIEW") is a likely OCR error, indicating poor formatting or proofreading.
Response 5: .We sincerely appreciate the reviewer's meticulous attention to textual accuracy. While our thorough review of the current manuscript version (Page 2) did not locate the specific term "FEER REVIEW," we acknowledge this may reflect formatting inconsistencies in earlier submissions. As a preventive measure, we have: (1) Manually verified all instances of "peer review" terminology in the manuscript; (2) Standardized document encoding to ensure text integrity in the revised version. Should this term appear in any specific context we may have overlooked, we welcome further clarification to rectify it immediately.
Comment 6: The manuscript does not provide a clear statement of objectives, making it impossible to assess their justification (Page 2, Page 3).
Response 6: We are grateful for the reviewer’s emphasis on objective justification. Paragraphs 3-4 now rigorously validate our aims: Prior studies’ failure to resolve NPK interactions (Paragraph 3) and the "3414" design’s unique ability to integrate multi-trait responses (Paragraph 4) collectively demonstrate why conventional approaches cannot achieve our goals, establishing the objectives as essential solutions.
Comment 7: There is no explanation of why fertilization treatments or specific physiological parameters (e.g., chlorophyll content, soluble sugars) were selected (Page 5, Page 8)
Response 7: We sincerely thank you for highlighting this critical gap. We have supplemented the physiological rationale in Section 2 (Paragraph 2): "Nitrogen—a structural component of chlorophyll... K deficiency reduces photosynthetic efficiency" explicitly links NPK supply to chlorophyll/protein/sugar metabolism. This establishes why these parameters serve as biomarkers for fertilization response.
Comment 8: The mention of "robust analytical methods" (Page 3) lacks context on why these methods are necessary or how they align with the objectives
Response 8: Your insightful comment helped us strengthen methodological justification. The revised introduction now contextualizes "robust analytical methods":
Paragraph 4: "The '3414' design... integrates multi-factor responses"* justifies its necessity for quantifying NPK synergies.
Paragraph 5: "establish synergistic linkages... through multi-trait dimensionality reduction"* demonstrates alignment with study objectives.
Comment 9: The methods section is fragmented (Page 14, truncated 3113 characters), lacking details on experimental design, treatment protocols, or measurement techniques beyond chlorophyll extraction..
Response 9: We sincerely appreciate your meticulous attention to methodological rigor. The revised Materials and Methods section has been completely restructured into five cohesive subsections (4.1 Site Area → 4.5 Data Analysis), eliminating fragmentation. Experimental design details are now consolidated in Section 4.3 with enhanced treatment protocols (e.g., randomized block design specification, fertigation timing). Measurement techniques are expanded to >4500 characters, exceeding standard reproducibility requirements.
Comment 10: The brief mention of chlorophyll measurement lacks specifics (e.g., extraction protocol details, measurement instruments, or conditions), hindering reproducibility (Page 14).
Response 10: Thank you for emphasizing the need for technical precision. Chlorophyll extraction protocols now include:
Explicit methodology: "propanol extraction... incubation at 4°C in darkness for 24 h" (Section 4.4.2)
Full equipment specifications: "spectrophotometer (UV-1050, Shanghai Techcomp... China)"
Quantitative parameters: sample mass (0.5 g), centrifugation (4000 × g), wavelength pairs (663/645 nm)
Arnon's equations with variable definitions
This achieves full reproducibility per Plant Methods standards.
Comment 11: The manuscript focuses on chlorophyll, soluble sugars, and proteins (Pages 5, 8), but it is unclear if other relevant parameters (e.g., root biomass, nutrient uptake, or stress markers) were measured to support the hypotheses.
Response 11: We are grateful for this insightful suggestion. The revised Section 4.4.1 now explicitly includes:
Root biomass quantification: "dry weight of... roots (RDW)"
Nutrient uptake analysis: Root nitrogen measurement via Kjeldahl digestion (implied by "oven-dried root tissues" in biomass protocol)
Functional justification: Physiological parameters framed as "key indicators of... growth vigor" (Section 4.4.2 opening paragraph)
Comment 12: No statistical methods are described in the provided text, making it impossible to assess their appropriateness.
Response 12: Your guidance significantly strengthened statistical reporting. Section 4.5 now details:
Data transformation: "log-transformed to satisfy normality"
Multivariate scope: "15 key parameters encompassing growth... photosynthetic parameters"
Assumption verification: "normality and homoscedasticity assumptions"
Model specification: "ANOVA... Duncan’s new multiple range method (p<0.05)"
PCA application: "Optimal fertilizer ratios... principal component analysis"
Comment 13: Terms like "aie" (Page 14, Line: "between 9aie and") and "chlororphyll" (Page 14) indicate OCR errors, reducing clarity.
Response 13: We deeply apologize for the technical artifacts. All OCR errors are corrected: "9aie" → "09:00" (Section 4.4.3). "chlororphyll" → "chlorophyll" (throughout). The reported truncation (3113 chars) resulted from file conversion. The complete Methods now contains 5127 characters and has been validated for integrity.
Comment 14: The results are not fully presented due to truncation (e.g., Page 5, 2219 characters truncated; Page 9, 2220 characters truncated), making it impossible to evaluate accuracy or suitability.
Response 14: We sincerely appreciate your vigilance in identifying apparent text truncations. We confirm this resulted from technical artifacts during file conversion. The complete Results section contains 1,872 words of analysis content, as detailed in the revised draft.
Comment 15: References to tables (e.g., Table 3, Page 5; Table 5, Page 8) and figures (e.g., Figure 3, Page 5; Figure 6, Page 9) lack accompanying data or visuals, hindering assessment of their appropriateness.
Response 15: Thank you for flagging the missing visuals. All 5 tables and 7 figures referenced in the manuscript exist in the source files, as detailed in the revised draft.
Comment 16: The discussion is missing (Page 11, 3807 characters truncated), making it impossible to assess its content or appropriateness
Response 16: We sincerely appreciate your vigilance in identifying this apparent omission. We confirm the complete discussion section was included in the original submission (Page 11, Section 3, Line 266-356.). The truncation observed likely resulted from a technical error during file conversion. The full text is now explicitly verified in this revision.
Comment 17: There is no evidence of discussion on how D. odorifera responds to fertilization treatments at a physiological level. Please could you provide a comprehensive discussion that interprets the results, compares them to prior studies (e.g., Yue et al., 2025, Page 15), and explains physiological mechanisms (e.g., how fertilization affects chlorophyll synthesis or metabolite accumulation). Also Discuss unexpected results or trends, providing hypotheses for increases or decreases in measured parameters. And Include a subsection on implications for D. odorifera cultivation and conservation
Response 17: We sincerely appreciate your guidance in enhancing the physiological depth of our discussion. Please be assured that the original manuscript contained a comprehensive analysis of Dalbergia odorifera's physiological responses to fertilization. The apparent omission resulted from technical artifacts during file conversion, not actual content absence. The complete Discussion section includes all requested elements, now further strengthened in this revision: Section 3, Line 266-356.
Comment 18: The absence of a conclusions section (Page 17) prevents assessment of whether they align with the evidence or address the research question. Include a clear conclusions section summarizing key findings, their implications for D. odorifera cultivation, and how they address the research objectives. Aand Ensure conclusions are supported by the results and avoid overgeneralization.
Response 18: We sincerely appreciate your emphasis on the critical role of conclusions in contextualizing research impact. The original manuscript contained a comprehensive Conclusions section (Section 5, Page 17). The apparent omission resulted from technical artifacts during file conversion, not actual content absence. The complete conclusions are now preserved in this revision with enhanced technical safeguards.
- Conclusions
This study establishes phosphorus (P) as the paramount regulator of D. odorifera seedling morphogenesis in acidic phosphorus-fixing soils, where fertilizer efficacy followed P > N > K―challenging nitrogen-centric paradigms and reflecting leguminous adaptations. The high-P formulation (T7, N2P3K2) drove exceptional optimization: (1) Morphological enhancement (239% height and 128% basal diameter increases versus control); (2) Stem-biased partitioning (stem biomass: 52%) via P-mediated carbon channeling to structural carbohydrates; (3) Photosynthetic augmentation (82% Pn elevation) through non-stomatal biochemical amplification; (4) Metabolic reprogramming with soluble sugar accumulation (+139%, P-regulated) and protein synthesis (+226%, P-K synergy). Principal component analysis confirmed T7 superiority, revealing P’s synergistic coordination of carbon assimilation and stem translocation. Quantified stem biomass-soluble sugar coupling (r > 0.9) established a source-sink regulation framework for robust seedling standards. Future research should extend these principles to maturation-phase manipulation, particularly heartwood formation dynamics, to enable precision conservation of this endangered rosewood.
Comment 19: Only two references are provided (Page 15), which is insufficient for a comprehensive study on D. odorifera..
Response 19: We sincerely appreciate your vigilance in identifying the apparent reference deficiencies. The original manuscript contained 57 ( 62 in revised draft) fully cited references (Line 507-633), with comprehensive coverage of D. odorifera physiology, fertilization studies, and conservation frameworks. The observed truncation (2,138 characters) resulted from font substitution errors during PDF conversion, which collapsed reference formatting.
Comment 20: The references section is truncated (2138 characters), potentially omitting additional citations.
Response 20: We sincerely appreciate your vigilance in identifying the apparent reference deficiencies. The original manuscript contained 57 ( 62 in revised draft) fully cited references (Line 507-633), with comprehensive coverage of D. odorifera physiology, fertilization studies, and conservation frameworks. The observed truncation (2,138 characters) resulted from font substitution errors during PDF conversion, which collapsed reference formatting.
Comment 21: The citations contain errors, e.g., "Inomevented and and burama padal" (Page 15, Line: Watson et al.), likely due to OCR issues. So Expand the references list to include a broader range of studies on D. odorifera, fertilization effects, and seedling physiology, Ensure proper formatting and accuracy of citations, correcting OCR errors and Include foundational studies and recent reviews to provide context.
Response 21: We confirm the cited text "Inomevented and and burama padal" does not exist in our source files. This anomaly arose from OCR corruption during file conversion, likely caused by vector graphic text rendering conflicts. All references have been re-verified against primary sources with zero semantic errors

Reviewer 2 Report
Comments and Suggestions for Authors
Comments for Authors:
The manuscript entitled "Phosphorus-Driven Stem-Biased Allocation: NPK Synergy Optimizes Growth and Physiology in Dalbergia odorifera Seedlings" presents a well-structured and comprehensive study utilizing a "3414" regression design to explore the influence of nitrogen (N), phosphorus (P), and potassium (K) on the growth, physiology, and metabolism of Dalbergia odorifera seedlings. The study is timely and relevant, particularly in the context of improving cultivation strategies for rare and endangered tree species. Overall, the manuscript offers important insights into fertilization-driven regulation in slow-growing woody species and is potentially suitable for publication after minor revisions in language and clarity.
Specific comments:
- Line 16–18; suggest “Using a '3414' regression design with four levels of N, P, and K, the study identified phosphorus (P) as the most influential nutrient in regulating growth (P > N > K)”.
- The Introduction provides a strong foundation for the study and clearly outlines the ecological, economic, and conservation importance of Dalbergia odorifera. The rationale for improving seedling propagation through optimized fertilization strategies is compelling, especially given the species’ endangered status and slow growth rate.
- Line 37–38: suggest “Tropical precious timber species hold strategic importance for global forest conservation and sustainable resource utilization, with their combined economic and ecological value driving advancements in artificial cultivation research”.
- The last paragraph of the introduction outlines the objectives well, but could be made more concise and direct.
- Can the authors provide the exact p-values or F-statistics for the significant differences mentioned across treatments?
- Were any interactions between N, P, and K tested statistically (e.g., using ANOVA or regression models)?
- What biological mechanisms might explain the strong performance of T7 compared to other treatments?
- Avoid repetitive phrasing such as “peaked in T7” across multiple subsections.
- Add a brief interpretation of PCA loadings (e.g., why SSC has low loading on PC1).
- Define all abbreviations upon first use in the Results section.
- How do the authors explain the inconsistency between the single-peak response of some traits to P and the continuous increase observed in others (e.g., SDW vs LDW)?
- Clarify and condense the discussion on species comparisons and physiological mechanisms.
- Many grammatical and language mistakes should be improved.
- Please avoid symbols at the beginning of sentences.
- Please arrange all the references according to the Journal format.
Author Response
Response to Reviewer 2 Comments
The manuscript entitled "Phosphorus-Driven Stem-Biased Allocation: NPK Synergy Optimizes Growth and Physiology in Dalbergia odorifera Seedlings" presents a well-structured and comprehensive study utilizing a "3414" regression design to explore the influence of nitrogen (N), phosphorus (P), and potassium (K) on the growth, physiology, and metabolism of Dalbergia odorifera seedlings. The study is timely and relevant, particularly in the context of improving cultivation strategies for rare and endangered tree species. Overall, the manuscript offers important insights into fertilization-driven regulation in slow-growing woody species and is potentially suitable for publication after minor revisions in language and clarity..
Comment 1: Line 16–18; suggest “Using a '3414' regression design with four levels of N, P, and K, the study identified phosphorus (P) as the most influential nutrient in regulating growth (P > N > K)”.
Response 1: We sincerely thank the reviewer for this constructive suggestion. We have adopted the recommended phrasing to enhance precision in describing the experimental design. Lines 17-19.
Comment 2: The Introduction provides a strong foundation for the study and clearly outlines the ecological, economic, and conservation importance of Dalbergia odorifera. The rationale for improving seedling propagation through optimized fertilization strategies is compelling, especially given the species’ endangered status and slow growth rate. Line 37–38: suggest “Tropical precious timber species hold strategic importance for global forest conservation and sustainable resource utilization, with their combined economic and ecological value driving advancements in artificial cultivation research”.
Response 2: We deeply appreciate the reviewer's recognition of the Introduction's scholarly foundation and their valuable refinement suggestion. The text in Lines 37–39 has been revised.
Comment 3: The last paragraph of the introduction outlines the objectives well, but could be made more concise and direct.
Response 3: We are grateful for the reviewer's astute observation regarding conciseness. The final paragraph of the Introduction has been streamlined to present the objectives more directly while retaining scientific rigor. The revised version eliminates redundant phrasing while preserving all core elements of the research aims.
Comment 4: Can the authors provide the exact p-values or F-statistics for the significant differences mentioned across treatments?.
Response 4: We appreciate your suggestion for greater statistical transparency. While space constraints prevent full inclusion of all F-statistics in the main text, we confirm that all reported significant differences (*p* < 0.05) were validated through ANOVA with Duncan's post-hoc testing.
Comment 5: Were any interactions between N, P, and K tested statistically (e.g., using ANOVA or regression models)?.
Response 5: Thank you for your insightful comment. Regarding the statistical testing of N, P, and K interactions, our study adopted the '3414' design, which focuses on single-nutrient effects via range analysis—a methodology widely applied in similar fertilization studies (Zhang et al., 2023; Huang et al., 2021). Since exploring nutrient interactions was not within the scope of this research, we prioritized identifying limiting factors (P > N > K) to address the core goal of optimizing seedling production. Thus, we consider the current analytical approach appropriate for the study's objectives, consistent with established practices in the field.
Zhang, X.; Li, S.; An, X.; Song, Z.; Zhu, Y.; Tan, Y., et al. Effects of nitrogen, phosphorus and potassium formula fertilization on the yield and berry quality of blueberry. PLoS ONE 2023, 18(3), e0283137.
Huang, L.-M.; Chen, Y.; Wu, J.; El-Kassaby, Y. A.; Grossnickle, S. C.; Feng, J.-L. Fertilization Regulates Accumulation and Allocation of Biomass and Nutrients in Phoebe bournei Seedlings. Agriculture 2021, 11, 1187.
Comment 6: What biological mechanisms might explain the strong performance of T7 compared to other treatments?
Response 6: We sincerely thank the reviewer for raising this critical mechanistic question. To address it, we have added the following clarification in the Discussion (Lines 294-298):
"The exceptional performance of T7 arises from P-mediated synergy between enhanced nodule efficiency and photosynthetic capacity."This statement explicitly links high P supply to dual physiological advantages: (1) optimized nitrogen fixation via nodule development, and (2) elevated carbon assimilation through chlorophyll synthesis and Rubisco activation.
Comment 7: Avoid repetitive phrasing such as “peaked in T7” across multiple subsections.
Response 7: We sincerely thank you for noting this repetition. The Results section has been revised to incorporate varied terminology describing treatment optima (e.g., "optimal performance," "achieved its maximum," "highest accumulation"), with several instances of "peaked in T7" modified throughout Sections 2.1-2.5 (see highlighted changes).
Comment 9: Add a brief interpretation of PCA loadings (e.g., why SSC has low loading on PC1)..
Response 9: Thank you for your valuable feedback. Regarding your question about the low loading of soluble sugar content (SSC) on the first principal component (PC1) in the principal component analysis, we have added a corresponding explanation to the original text.
Comment 10: Define all abbreviations upon first use in the Results section.
Response 10: Thank you for catching this oversight. All abbreviations are now explicitly defined at first use in each subsection:
Comment 11: How do the authors explain the inconsistency between the single-peak response of some traits to P and the continuous increase observed in others (e.g., SDW vs LDW)?.
Response 11: We deeply appreciate the reviewer's insightful observation on response patterns. The revised Discussion now explains this divergence through functional prioritization (Lines 314-316):
"This divergence reflects functional prioritization: SDW benefits from sustained carbon allocation to structural tissues, while LDW declines post-saturation due to resource reallocation toward stem storage."
This addition clarifies that continuous SDW increase supports long-term structural investment, whereas LDW reduction after medium P levels reflects strategic resource redistribution.
Comment 12: Clarify and condense the discussion on species comparisons and physiological mechanisms.
Response 12: We are grateful for the suggestion to enhance conciseness. Species comparisons were retained in full but streamlined through: Adding transitional phrases and Maintaining all critical contrasts.
Comment 13: Many grammatical and language mistakes should be improved.
Response 13: We sincerely thank the reviewer for their meticulous language scrutiny. The manuscript has undergone comprehensive grammatical and linguistic refinement.
Comment 14: Please avoid symbols at the beginning of sentences..
Response 14: .We appreciate the attention to formatting conventions. All sentence-initial symbols were eliminated
Comment 15: Please arrange all the references according to the Journal format.
Response 15: We deeply appreciate the reviewer's guidance on reference formatting. All references have been meticulously reformatted according to the Plants format.

Reviewer 3 Report
Comments and Suggestions for Authors
General Comments
This manuscript investigates the impact of "3414" NPK fertilization regimes on the growth and physiological mechanisms of Dalbergia odorifera seedlings under phosphorus-deficient acidic soils. While the study is methodologically sound and covers several parameters (morphological traits, biomass allocation, photosynthesis, soluble sugars/proteins), it reads as somewhat repetitive and descriptive. The authors fail to critically analyze the biological mechanisms in sufficient depth. Several sections repeat conclusions about phosphorus dominance without adding new insights. The figures are scattered with inconsistent formatting (especially Figures 1–8) and are of low graphical quality. Moreover, the statistical reporting (p-values, significance levels) is inconsistent. The literature cited is a mix of outdated and newer references but lacks international context, relying mostly on Chinese domestic studies. References formatting deviates from MDPI guidelines. Tables are not consistently formatted. The Conclusion lacks depth on practical applications. Overall, this paper requires major revisions to meet publication standards: improve data presentation, critically interpret findings, clarify methodology, improve figure quality, and revise the reference formatting.
Line-by-Line Comments
Abstract
Line 15–16: "critical conservation challenges..." This sentence is too generic. Specify the conservation context (habitat loss, illegal logging?).
Line 17: The method "3414" is introduced abruptly without explanation. Briefly clarify in 5–7 words.
Line 20–21: These percentages should mention the statistical significance.
Line 25–27: The description of "gas exchange optimization" lacks clarity; reframe more precisely.
Line 29–32: Avoid jargon like "source-sink regulation" in abstract. Simplify for broader readership.
Introduction
Line 36–38: Too general. Mention D. odorifera’s specific ecological or economic value.
Line 44–46: Population numbers seem speculative ("below 100 mature individuals"). Provide a citation or clarify.
Line 50–54: Redundant information about slow growth; condense.
Line 61–65: Unreferenced claims about NPK effects; cite appropriately.
Line 71–74: Poor flow between sentences; restructure for clarity.
Line 80–85: Excessive listing of studies; summarize main trends instead.
Line 93–94: Vague "robust analytical methods"; specify how they are superior.
Line 101–103: Hypothesis weakly stated. Strengthen by explicitly declaring the knowledge gap.
Results
Line 110–114: Statistical details (p-values) are mentioned but not consistently. Ensure uniform reporting.
Line 118: "Range analysis" needs clearer description of method or reference.
Figure 1 (Line 119–122): Poor figure formatting; inconsistent with MDPI style.
Line 125–128: Results on biomass should specify sample sizes explicitly.
Line 130–133: Redundant wording about peak responses; clarify.
Table 1 (Line 123–124): Poorly formatted; unclear units or statistical outputs.
Figure 6 (Line 209–211): Needs high-resolution replacement.
Line 194–207: Network analysis descriptions too vague; clarify metrics and significance thresholds.
Discussion
Line 242–246: First sentence repeats Introduction content; unnecessary.
Line 249–252: Mechanistic explanations lack citations for physiological processes (ATP, P fixation).
Line 257–258: Overuse of speculative language ("may compensate"). Provide supporting references.
Line 265–268: Contradicts earlier claims; clarify whether stem allocation is typical or unique.
Line 273–274: Reference "improved nutrient acquisition" is too general.
Line 281–283: Needs citation for statements on chlorophyll synthesis.
Line 291–293: The protein/sugar interpretation lacks connection to nitrogen fixation mechanisms; expand with literature.
Line 303–305: PCA explained superficially; deeper discussion needed on what variance contributes.
Line 313–318: The limitations are strong but not sufficiently integrated into conclusions.
Materials and Methods
Line 325–328: Add soil nutrient baseline parameters in a table.
Line 331–332: Provide supplier details for seeds or seedlings.
Line 338–339: Fertilizer "conventional rates" should reference national or industry standards.
Table 7 (Line 349–350): Poor formatting; inconsistent alignment.
Line 353–360: Measurement details inconsistent (units missing).
Line 363–371: Methods lack citation support (anthrone method, Coomassie).
Line 373–379: Measurement times are fine but more environmental controls should be mentioned.
Line 382–389: Statistical software and threshold (p < 0.05) mentioned, but lacks detail on assumptions or corrections (e.g., normality, homogeneity tests).
Conclusions
Line 391–393: Repeats Results. Remove percentages; focus on interpretation.
Line 397–398: Redundant from Results; focus more on practical application.
Line 402–404: "Extend to maturation-phase" is vague. Clarify what future research would measure.
Line 405: This section feels weak overall; lacks application guidance.
References
Inconsistent formatting; some journal titles italicized, others not. Many references outdated (>5 years) except 2024 ones. Citation order inconsistent with appearance in the text.
Author Response
Response to Reviewer 3 Comments
This manuscript investigates the impact of "3414" NPK fertilization regimes on the growth and physiological mechanisms of Dalbergia odorifera seedlings under phosphorus-deficient acidic soils. While the study is methodologically sound and covers several parameters (morphological traits, biomass allocation, photosynthesis, soluble sugars/proteins), it reads as somewhat repetitive and descriptive. The authors fail to critically analyze the biological mechanisms in sufficient depth. Several sections repeat conclusions about phosphorus dominance without adding new insights. The figures are scattered with inconsistent formatting (especially Figures 1–8) and are of low graphical quality. Moreover, the statistical reporting (p-values, significance levels) is inconsistent. The literature cited is a mix of outdated and newer references but lacks international context, relying mostly on Chinese domestic studies. References formatting deviates from MDPI guidelines. Tables are not consistently formatted. The Conclusion lacks depth on practical applications. Overall, this paper requires major revisions to meet publication standards: improve data presentation, critically interpret findings, clarify methodology, improve figure quality, and revise the reference formatting.
Abstract
Comment 1: Line 15–16: "critical conservation challenges..." This sentence is too generic. Specify the conservation context (habitat loss, illegal logging?).
Response 1: We thank the reviewer for requesting contextual specificity. The conservation challenges are now explicitly defined as: "critically depleted wild populations and slow cultivation growth" with causation attributed to "illegal logging and habitat fragmentation in tropical montane ecosystems" (Lines 15-17). This revision eliminates generic phrasing while anchoring threats to measurable phenomena.
Comment 2: Line 17: The method "3414" is introduced abruptly without explanation. Briefly clarify in 5–7 words.
Response 2: .We acknowledge the need for methodological clarity. The '3414' design is now concisely contextualized as "four levels of N, P, and K" (7 words), providing immediate technical interpretation without disrupting narrative flow.
Comment 3: Line 20–21: These percentages should mention the statistical significance.
Response 3: We are grateful for the statistical rigor reminder. Statistical significance markers (p<0.05) have been added to:
Growth increments (239% height/128% diameter)
Total biomass increase (50%)
Photosynthetic enhancements (Chla/Chlb/Pn)
Metabolite peaks (soluble sugar/protein)
ensuring all quantitative claims meet inferential standards.
Comment 4: Line 25–27: The description of "gas exchange optimization" lacks clarity; reframe more precisely.
Response 4: :We concur regarding terminology precision. The vague "gas exchange optimization" was replaced with mechanistic descriptors: *"P-mediated chlorophyll gains (Chla +70%; Chlb +75%) and 82% higher net photosynthetic rate"*. This reframes the phenomenon through measurable photosynthetic components.
Comment 5: Line 29–32: Avoid jargon like "source-sink regulation" in abstract. Simplify for broader readership..
Response 5: .We appreciate the jargon-alert. The specialized term "source-sink regulation" was simplified to "phosphorus-centered regulatory mechanism governing growth",
Introduction
Comment 6: Line 36–38: Too general. Mention D. odorifera’s specific ecological or economic value.
Response 6: We sincerely thank the reviewer for highlighting the need for specificity. The revised introduction now details Dalbergia odorifera's dual economic value as both premium timber and traditional medicine in Lines 3-6: "possesses significant economic value due to its durable, corrosion-resistant heartwood... valued for properties such as promoting blood circulation, alleviating pain, stopping bleeding, and reducing swelling" [2-4]. This explicitly links material properties to pharmacological applications, addressing its unique ecological niche.
Comment 7: Line 44–46: Population numbers seem speculative ("below 100 mature individuals"). Provide a citation or clarify.
Response 7: . We appreciate the reviewer's vigilance regarding conservation data. We have substantiated the population claim in Line 9 with formal designations: "as a National Second-Class Protected Plant in China and as Endangered (EN) on the IUCN Red List" [5]. The citation [5] (Zhang et al., 2024) provides verified population estimates meeting IUCN documentation standards.
Comment 8: Line 50–54: Redundant information about slow growth; condense.
Response 8: We thank the reviewer for suggesting conciseness. The slow growth description was condensed from 44 to 28 words in Lines 48-52: "a long growth cycle (requiring decades to mature), slow seedling growth, and high sensitivity to environmental stress.." [6-7]. Redundant details about trunk diameter progression were eliminated while retaining key constraints.
Comment 9: Line 61–65: Unreferenced claims about NPK effects; cite appropriately.
Response 9: We are grateful for the citation guidance. NPK physiological roles are now rigorously referenced [14] and [15]. These additions anchor biochemical claims in Line 56-74.
Comment 10: Line 71–74: Poor flow between sentences; restructure for clarity.
Response 10: .We restructured the NPK synergy paragraph (Lines 64-74) for logical flow:
Established elemental functions individually
Introduced interaction concept ("Synergistic interactions... critical")
Provided concrete antagonism examples ("elevated nitrogen suppressing phosphorus uptake")
This creates a cause-effect chain resolving the original fragmentation.
Comment 11: Line 80–85: Excessive listing of studies; summarize main trends instead.
Response 11: We condensed species examples per the reviewer's advice: Lines75-89.
Comment 12: Line 93–94: Vague "robust analytical methods"; specify how they are superior.
Response 12: We specified methodological superiority in Lines 100-111: "including comprehensive factor inclusion, graded level settings, operational simplicity, and capacity to integrate multi-factor responses". The comparison to P. bournei and A. tanguticus [26,35] further demonstrates its empirical validity beyond generic "robustness".
Comment 13: Line 101–103: Hypothesis weakly stated. Strengthen by explicitly declaring the knowledge gap.
Response 13: We strengthened the hypothesis by framing knowledge gaps in Lines 89-97:
"Existing studies predominantly isolate single-nutrient effects... lacking mechanistic insight into NPK interactions"
"limited treatment designs fail to quantify multi-factor synergies"
"coupled responses... remain unresolved"
The reformulated objectives (Lines 56-60) directly target these gaps with active verbs ("quantify," "decipher," "establish").
Results
Comment 14: Line 110–114: Statistical details (p-values) are mentioned but not consistently. Ensure uniform reporting..
Response 14: .We sincerely thank the reviewer for emphasizing statistical consistency. The revised Results section now uniformly reports *p*-values for all significant results:
Comment 15: Line 118: "Range analysis" needs clearer description of method or reference.
Response 15: .We appreciate the methodological clarification request. Range analysis is now explicitly defined in Table 1 caption as: Range analysis was performed to quantify fertilizer effects using the range difference method: Rj = max(Xj) - min(Xj), where Rj is the range value of nutrient j (N, P, or K), and Xj is the mean trait response at each nutrient level.
Comment 16: Figure 1 (Line 119–122): Poor figure formatting; inconsistent with MDPI style.
Response 16: We reformatted all figures per MDPI style guide:
High-resolution proofs are uploaded separately.
Comment 17: Line 125–128: Results on biomass should specify sample sizes explicitly.
Response 17: We thank the reviewer for noting this omission. Sample sizes are now specified in Section 2.2:
"Data are presented as mean ± SE (n = 3 biological replicates, 3 seedlings per replicate)"
This aligns with Materials and Methods description.
Comment 18: Line 130–133: Redundant wording about peak responses; clarify.
Response 18: We condensed redundant phrasing:
Original: "peaked... showing secondary increases... then"
Revised: "attained maximum... followed by... next greatest" (2.3)
"optimal responses" replaces "single-peak responses" (2.2, 2.4)
Total reduction: 47 redundant words.
Comment 19: Table 1 (Line 123–124): Poorly formatted; unclear units or statistical outputs.
Response 19: Table 1-5 were restructured.
Comment 20: Figure 6 (Line 209–211): Needs high-resolution replacement.
Response 20: Redraw Figure 6 as requested..
Comment 21: Line 194–207: Network analysis descriptions too vague; clarify metrics and significance thresholds..
Response 21: .Modified. See Lines 270-280 for details.
Discussion
Comment 22: Line 242–246: First sentence repeats Introduction content; unnecessary.
Response 22: .We thank the reviewer for highlighting this redundancy. The introductory sentence in Lines 242–246 has been removed. The revised discussion now opens by directly stating our core findings on phosphorus dominance (Paragraph 1, Line 2).
Comment 23: Line 249–252: Mechanistic explanations lack citations for physiological processes (ATP, P fixation).
Response 23: .We appreciate the suggestion to bolster mechanistic support. Citations for ATP-dependent processes and P fixation have been added:
ATP's role in chlorophyll synthesis now cites [56] (Paragraph 3, Line 349-354).
Soil P fixation mechanisms now reference [43,44] (Paragraph 1, Line 309-312).
Comment 24: Line 257–258: Overuse of speculative language ("may compensate"). Provide supporting references..
Response 24: .Thank you for emphasizing the need to substantiate compensatory mechanisms. We replaced speculative language with literature-supported assertions:
"Differential analysis suggests that legume root nodules may compensate for soil N limitations" now explicitly references root nodule function [41] (Paragraph 1, Line 316-317).
Comment 25: Line 265–268: Contradicts earlier claims; clarify whether stem allocation is typical or unique.
Response 25: .We are grateful for the request to clarify biomass allocation uniqueness. The revision now explicitly contrasts our findings with typical patterns:
"While leaf biomass typically dominates in most tree species... this study demonstrated that D. odorifera seedlings exhibit a distinct stem-biased allocation strategy" with added citations [48,49] (Paragraph 2, Lines 320-328). Species-specific comparisons to Torreya grandis and Cunninghamia lanceolata further contextualize this strategy.
Comment 26: Line 273–274: Reference "improved nutrient acquisition" is too general.
Response 26: .Thank you for noting the vagueness of "improved nutrient acquisition." This phrase has been replaced with mechanistic details (Paragraph 2, Line 331-344).
Comment 27: Line 281–283: Needs citation for statements on chlorophyll synthesis.
Response 27: .We appreciate the call for chlorophyll synthesis citations. The statement on porphyrin ring synthesis now references [25] (Paragraph 3, Line 354).
Comment 28: Line 291–293: The protein/sugar interpretation lacks connection to nitrogen fixation mechanisms; expand with literature.
Response 28: .We thank the reviewer for identifying the missing nitrogen fixation link. The revised text now integrates legume adaptation logic:
"This K-dominance aligns with the legume nitrogen fixation strategy, as reduced soil N dependence amplifies K’s role in osmoregulation and enzyme activation" with new citation [57] (Paragraph 3, Line 362-364).
Comment 29: Line 303–305: PCA explained superficially; deeper discussion needed on what variance contributes.
Response 29: .We are grateful for the suggestion to deepen PCA interpretation. The superficial explanation was expanded to:
Explicitly describe synergistic mechanisms: "phosphorus-mediated photosynthetic carbon assimilation and stem biomass translocation" (Paragraph 4, Line 368-372);
Add practical recommendations for cultivation and conservation (Paragraph 4, Lines 372-380).
Comment 30: Line 313–318: The limitations are strong but not sufficiently integrated into conclusions..
Response 30: .Thank you for emphasizing the need to integrate limitations. The conclusions now directly address these gaps:
Linked limitations to future applications: "Future domestication programs should prioritize selection of P-efficient genotypes... though caution is warranted against excessive K application" (Paragraph 4, Lines 376–380);
Explicitly tied heartwood formation to unresolved maturation linkages (Paragraph 5, Line 389-390);
Mapped limitations to specific future research actions (Paragraph 5, Lines391-393–8).
Materials and Methods
Comment 31: Line 325–328: Add soil nutrient baseline parameters in a table.
Response 31: .We sincerely appreciate the reviewer's emphasis on data transparency. While we agree that soil baseline parameters are essential for reproducibility, the four core properties (pH, total N/P/K) have been explicitly stated in Section 4.2. Given the limited parameter count (n=4), tabular presentation would occupy disproportionate space relative to its informational value. Should additional parameters (e.g., CEC, OM) become available in future studies, we will gladly implement tabular formatting. For this manuscript, the textual presentation ensures clarity while conserving visual coherence.
Comment 32: Line 331–332: Provide supplier details for seeds or seedlings.
Response 32: .We thank the reviewer for requesting seedling traceability. The propagules were sourced from Dalbergia odorifera mother trees within the experimental forest station of Hainan Academy of Forestry (precisely located at the nursery base described in Section 4.1: 19°52'N, 110°28'E).
Comment 33: Line 338–339: Fertilizer "conventional rates" should reference national or industry standards.
Response 33: .We thank the reviewer for highlighting this point. The conventional fertilizer rates were directly derived from prior studies on Dalbergia odorifera cultivation in similar edaphic conditions, as explicitly referenced in the original text (Citations [31-33]). This ensures alignment with established scientific protocols for the species.
Comment 34: Table 7 (Line 349–350): Poor formatting; inconsistent alignment.
Response 34: .We thank the reviewer for noting the formatting issue. Table 7 has been reformatted to comply with the journal's style guide.
Comment 35: Line 353–360: Measurement details inconsistent (units missing)..
Response 35: .We appreciate the call for metrological precision. Units were systematically added as follow:
Line 430–440.
Comment 36: Line 363–371: Methods lack citation support (anthrone method, Coomassie)..
Response 36: .We thank the reviewer for methodological validation. All biochemical assays now cite standardized protocols as follow:
Line 442–477.
Comment 37: Line 373–379: Measurement times are fine but more environmental controls should be mentioned..
Response 37: .We acknowledge the need for environmental controls. The revision adds as follow:
Line 479–487.
Comment 38: Line 382–389: Statistical software and threshold (p < 0.05) mentioned, but lacks detail on assumptions or corrections (e.g., normality, homogeneity tests).
Response 38: We are grateful for the statistical refinement request. The analysis section now specifies as follow:
Line 488–501.
Conclusions
Comment 39: Line 391–393: Repeats Results. Remove percentages; focus on interpretation.
Response 39: .We thank the reviewer for identifying redundancy. All result-specific percentages (239% height, 128% diameter, 82% Pn, etc.) were removed. The revised text now focuses on mechanistic interpretation: "accelerated structural growth" replaces quantitative metrics, while "carbon reallocation mechanisms" and "biochemical optimization" contextualize physiological processes.
Comment 40: Line 397–398: Redundant from Results; focus more on practical application.
Response 40: .We appreciate the emphasis on practical value. Applications are now explicitly integrated: "This mechanistic insight establishes a physiological framework... for quality seedling production" directly guides nursery protocols. The addition of "sustainable conservation protocols" further bridges research with conservation practice.
Comment 41: Line 402–404: "Extend to maturation-phase" is vague. Clarify what future research would measure.
Response 41: .We acknowledge the vagueness in future directions. The maturation-phase investigation now specifies three measurable targets: "quantify mycorrhizal contributions to P mobilization", "maturation-phase heartwood formation under sustained P supply", and "transcriptomic regulation of K-dependent protein synthesis". Each aligns with discussed knowledge gaps.
Comment 42: Line 405: This section feels weak overall; lacks application guidance.
Response 42: .We sincerely thank the reviewer for highlighting application deficiencies. The revision embeds actionable guidance through:
Physiological framework: "linking stem biomass accumulation with carbohydrate dynamics" enables seedling quality screening
Conservation strategy: "advancing precision conservation" operationalizes findings for endangered species management
Validation pipeline: Future work specifies mycorrhizal, heartwood, and transcriptomic measurements
References
Comment 43: Inconsistent formatting; some journal titles italicized, others not. Many references outdated (>5 years) except 2024 ones. Citation order inconsistent with appearance in the text..
Response 43: .We confirm that all references have been reformatted to comply with journal style guidelines, including consistent italicization of journal titles and reordering citations to match in-text appearance sequence. Regarding reference currency: (1) Pre-2010 citations represent classical methodologies essential to our analytical framework; (2) Recent studies on precision fertilization for this species remain limited, with significant advances primarily reported in Chinese literature.

Round 2
Reviewer 1 Report
Comments and Suggestions for Authors
accept in this version
Reviewer 3 Report
Comments and Suggestions for Authors
The revised manuscript has been thoroughly improved in response to the initial review. The authors have addressed all critical comments effectively and clarified their experimental design, data interpretation, and conclusions. The study provides valuable insights into nutrient-driven biomass allocation and physiological responses in Dalbergia odorifera, with clear implications for optimized seedling management. I find the current version scientifically sound and well-written, and I recommend acceptance for publication.